# Weight trajectories in aging humanized APOE mice with translational validity to human Alzheimer's risk population: A retrospective analysis

Francesca Vitali [1,2,3,4] *, Jean-Paul Wiegand[1], Lillian Parker-Halstead[1], Allan Tucker[5], Roberta Diaz Brinton[1,2,3,4,6]

1 Center for Innovation in Brain Science, University of Arizona Health Sciences, Tucson, Arizona, United States of America, 2 Department of Neurology, College of Medicine Tucson, University of Arizona, Tucson, Arizona, United States of America, 3 BIO5 Institute, University of Arizona, Tucson, Arizona, United States of America, 4 Center for Biomedical Informatics and Biostatistics, Tucson, Arizona, United States of America, 5 Department of Computer Science, Brunel University London, Uxbridge, United Kingdom, 6 Department of Pharmacology, College of Medicine Tucson, University of Arizona, Tucson, Arizona, United States of America

* francescavitali@arizona.edu

**Data Availability Statement:** All relevant data are within the manuscript and its Supporting Information files. Python code along with data are

## Abstract

Translational validity of mouse models of Alzheimer's disease (AD) is variable. Because change in weight is a well-documented precursor of AD, we investigated whether diversity of human AD risk weight phenotypes was evident in a longitudinally characterized cohort of 1,196 female and male humanized APOE (hAPOE) mice, monitored up to 28 months of age which is equivalent to 81 human years. Autoregressive Hidden Markov Model (AHMM) incorporating age, sex, and APOE genotype was employed to identify emergent weight trajectories and phenotypes. In the hAPOE-AD mouse cohort, five distinct weight trajectories emerged: three trajectories were associated with a weight loss phenotype (36% of mice, n = 426), one with weight gain (13% of mice, n = 152), and one trajectory of no change in weight (34% of mice, n = 403). The AHMM model findings were validated with post-hoc survival analyses, revealing differences in survival rates across the five identified phenotypes. Further validation was performed using body composition and plasma β-amyloid data from mice within the identified gain, loss and stable weight trajectories. Weight gain trajectory was associated with elevated plasma β-amyloid levels, higher body fat composition, lower survival rates and a greater proportion of APOE4/4 carriers. In contrast, weight loss was associated with greater proportion of hAPOE3/4 carriers, better survival rates and was predominantly male. The association between weight change and AD risk observed in humans was mirrored in the hAPOE-AD mouse model. Weight trajectories of APOE3/3 mice were equally distributed across weight gain, loss and stability. Surprisingly, despite genetic uniformity, comparable housing, diet and handling, distinct weight trajectories and divergence points emerged for subpopulations. These data are consistent with the heterogeneity observed in the human population for change in body weight during aging and highlight the importance of longitudinal phenotypic characterization of mouse aging to advance the translational validity of preclinical AD mouse models.

available at https://github.com/fransiss/AHMM_mouse_trajectories.

**Funding:** Research reported herein was supported by the National Institute on Aging (grants P01AG026572 [Perimenopause in Brain Aging and Alzheimer's Disease], T32AG061897 [Translational Research in Alzheimer's Disease and Related Dementias (TRADD)], 5R01AG057931-02 [Sex Differences in Molecular Dementias of Alzheimer's Disease Risk: Prodromal Endophenotype]), the Women's Alzheimer's Movement to Roberta Diaz Brinton, and the University of Arizona Center for Innovation in Brain Science. The funders had no role in study design, data collection and analysis, decision to publish, or preparation of the manuscript.

**Competing interests:** The authors have declared that no competing interests exist.

## Introduction

Alzheimer's disease (AD) is a multifactorial and complex brain disease resulting from multiplicative combinations of risk factors including sex, age, genetics, lifestyle, or environmental factors [1]. AD is currently the fifth-leading cause of death among Americans aged 65 or older, with approximately one in nine individuals in this age group diagnosed with dementia due to AD [1,2].

Recently, FDA-approved amyloid-targeting monoclonal antibodies show promise in slowing cognitive decline during the early stages of AD [3,4]. However, amyloid-targeting monoclonal antibodies do not prevent or reverse the disease and are associated with significant safety concerns, high costs, and little to no efficacy in women [5]. This underscores the need for more comprehensive approaches to prevent and treat AD, given its complex etiology.

Although late-onset AD is typically diagnosed at age 65 or older, the pathological changes and biological mechanisms driving the disease begin 20 years or more prior to AD diagnosis, during the prodromal/preclinical phase of AD [1]. The preclinical phase of AD is a critical window for intervention.

A fundamental requirement of preclinical models is the transitional validity to inform fundamental mechanistic biology for successful therapeutic development. Validating mouse models by identifying consistent AD precursors in humans is a possible approach for advancing translational validity. Robust animal models that accurately mimic the complexity of AD are essential for selecting the right population for clinical trials [6–8].

A possible strategy for potential reverse translational validity of mouse models is to establish associations between human observations and their manifestation in animal models [9–11]. Herein, we focused our analysis on the well-documented association between a weight change and AD risk, as observed in humans [12–21]. Elevated body mass index, weight increase, and obesity have been linked with a higher risk of AD decades prior to diagnosis [19,20], whereas weight loss is more proximate to AD diagnosis by 1 to 3 years [16,18,21]. Moreover, these associations are influenced by sex, with weight change more strongly associated with increased risk in women [15]. This highlights the diversity of human-based weight phenotypes, which might exacerbate AD risk factors, including sex, comorbidities and the major genetic risk factor APOE genotype [18,22–24].

In this study, we sought to determine whether an aging population of Model Organism Development and Evaluation for Late-Onset Alzheimer's Disease (MODEL-AD) [25] humanized APOE (hAPOE) mouse colony mirrored the changes in weight observed in the human aging population. hAPOE mouse models are established models to study the functional roles of human APOE genotypes in AD-related processes [26]. The importance of considering the impact of biological variables such as age, sex and human APOE genotypes on disease processes has been well-established for translational validity of mouse models of AD risk [26]. In this study, we focus on the multivariate nature of aging in the context of sex and APOE genotype biology which is relevant to the vast majority of individuals at risk for developing AD and therefore relevant to development of therapeutics to prevent, delay and treat late onset AD.

Longitudinal observations of weight data spanning from 5 to 28 months of age within an aged hAPOE mouse colony of 1,196 hAPOE mice (Fig 1) were analyzed to investigate if distinct weight trajectories exist. The mouse colony was composed of both male and female transgenic C57BL/6J mice carrying different hAPOE alleles (hAPOE3/3, hAPOE3/4, or hAPOE4/4) and thus the influence of sex and APOE genotype could be determined in the weight trajectories.

To determine the trajectories of weight change across age, we employed an Autoregressive Hidden Markov Model (AHMM) (Fig 2), a probabilistic approach that effectively handles

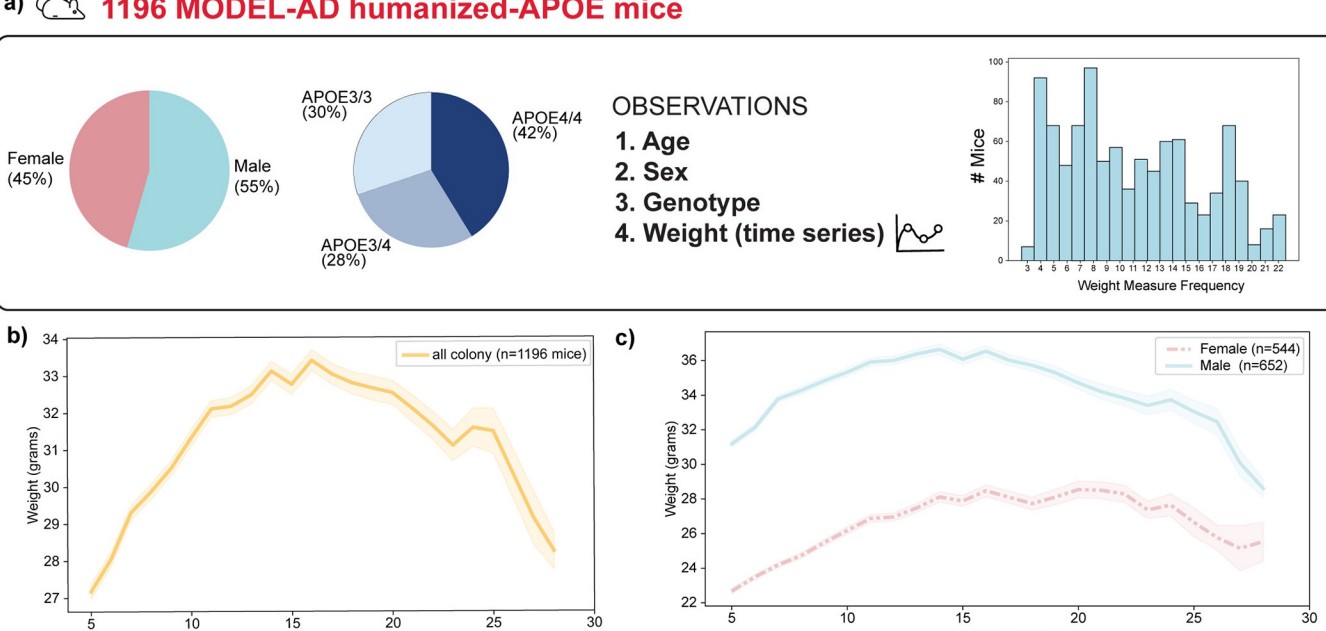

**Fig 1. Model Organism Development and Evaluation for Late-Onset Alzheimer's Disease (MODEL-AD) humanized APOE (hAPOE) colony. a)** an overview of the mouse colony is presented, displaying the percentages of sex and APOE genotype, along with the variables included in the model and the frequency of weight measurements. **b)** illustrates the mean and standard deviation of longitudinal weights of the mice colony. **c)** shows the mean and standard deviation of weights by sex, with different colors and line styles representing sexes.

longitudinal data collection at different and not-equally spaced time points, and missing data. In literature, AHMMs have been applied to speech signal characterization [27], and to early detection of neonatal sepsis [28], among others [29,30]. We applied an adapted version of AHMM to identify distinct groups of mice following the same weight trajectories.

The proposed model and the identified trajectories were validated using post-hoc survival data and analysis of independent measurements of plasma β-amyloid (Aβ) concentration and EchoMRI observations of body composition. Additionally, we analyzed the distribution of sex and APOE genotypes across trajectories to assess differences among the identified groups. The outcomes of these analyses revealed a diversity of weight trajectories consistent with the diversity observed in human population.

## Materials and methods

### Weight data and preprocessing

Weight data were collected for 2,280 hAPOE3/3, hAPOE3/4 and hAPOE4/4 mice from November 2nd 2018 to January 28th 2022. Monthly weight of the animals was measured with a monthly calibrated Ohaus SPX422 with a precision of 0.01g (Ohaus Precision Balance; Hogentogler & Co. Inc., Columbia, MD). Mice with a monthly weight loss exceeding 20% were excluded from the study. Weight measurements were collected at varying ages and frequencies, resulting in weight sequences that differ in length and spacing.

Initial preprocessing of weight data involved exclusion of breeders due to their diet change upon mating (from Teklad 7913 to Teklad Global 2919 breeder diets), which could directly impact their weight. Additionally, data were pre-processed to exclude collection errors, such as instances where weight collection date occurred either after the date of death or before the

date of birth. Weight measurements were included for mice of at least 5 months of age and those that were weighed for at least 3 time points. The distribution of the weight collection frequency is reported in Fig 1A. This preprocessing analysis resulted in the inclusion of 1,196 mice in the study.

**Mice study cohort.** Animal studies were performed following National Institutes of Health guidelines on use of laboratory animals. The protocol (16–170) was approved by the University of Arizona Institutional Animal Care and Use Committee (IACUC), and this study is reported in according with ARRIVE guidelines [31,32]. Breeder pairs were obtained from the Jackson Laboratory: B6(SJL)-APOE$^{tm1.1(APOE^*4)Adiuj\ /J}$ (Stock No: 027894, humanized APOE4 KI) and B6.Cg-Apoe$^{em2(APOE^*)Adiuj\ /J}$ (Stock No: 029018, humanized APOE3 KI). Animals were inbred to generate colonies of each line, and cross-bred to generate an APOE3/4 KI. The colonies were generated and maintained in a health status A facility on 14-h:10-h light: dark cycle and housed in ventilated cages with corn cob bedding and a single nestlet, provided with irradiated Envigo NIH-31 diet 7913 and sterile water ad libitum, and a Sheperd Shack® [33] enrichment house when single-housed. Upon observation of adverse events, University Animal Care Veterinary Services provided treatment until euthanasia (via cervical dislocation or carbon dioxide overdose) was deemed necessary. Studies involving interventions beyond behavioral characterizations were excluded from the analyses due to the multi-purpose use of this mouse cohort for various experiments and research projects.

## Autoregressive hidden Markov models

To identify different heterogeneous aging profiles of the hAPOE mice colony, i.e. trajectories, we utilized a computational strategy based on a variation of the Hidden Markov Models (HMM).

HMMs are probabilistic models designed to unveil hidden states that govern sequential patterns and the probability of their occurrence within a set of longitudinal observations. These models allow the identification of trajectories and phenotypes that more likely occurs and can describe the longitudinal measurements.

In this study, the observations consist of longitudinal weight data of the mice colony, along with the related mouse age at data collection, sex and APOE genotype. HMMs assume that these observations are generated by an unobserved sequence of states $H = \{H_1, H_2, \ldots, H_N\}$, referred as hidden states (Fig 2). These states capture latent conditions or patterns that influence the observed weights at different time points. Hidden states can assume N possible discrete values and N is selected empirically and usually based on the application and the data. For this study, N was set to 10 and named alphabetically as A, B, C, D, E, F, G, H, I, and J.

HMMs assume that the transitions between hidden states follow a stochastic process of a Markov chain (Fig 2) and these transitions are represented by the transition probability matrix $A = [a_{i,j}]$. Each element $a_{i,j}$ of the matrix denotes the probability of transitioning from the hidden state $i$ to the hidden state $j$ at time $t$, given the hidden state was $i$ at time $t$-1. These probabilities can be considered to unveil different temporal patterns corresponding to different trajectories and can be represented with a diagram (Fig 2) for a better interpretation of the transition patterns existing between the different hidden states (trajectories).

In this study, we applied an Autoregressive Hidden Markov Model (AHMM) that, unlike standard HMMs where the current observation is independent from all the other observations, is also autoregressive introducing direct stochastic dependencies in the sequential data. In the case of AHMM, each observation (mouse) corresponds to an autoregressive time series $X_t = \{X_1, X_2, \ldots, X_k\}$ with $k$ components (weight measurements) where observations from prior time points inform predictions at the next time step. This allows to account for the dependency

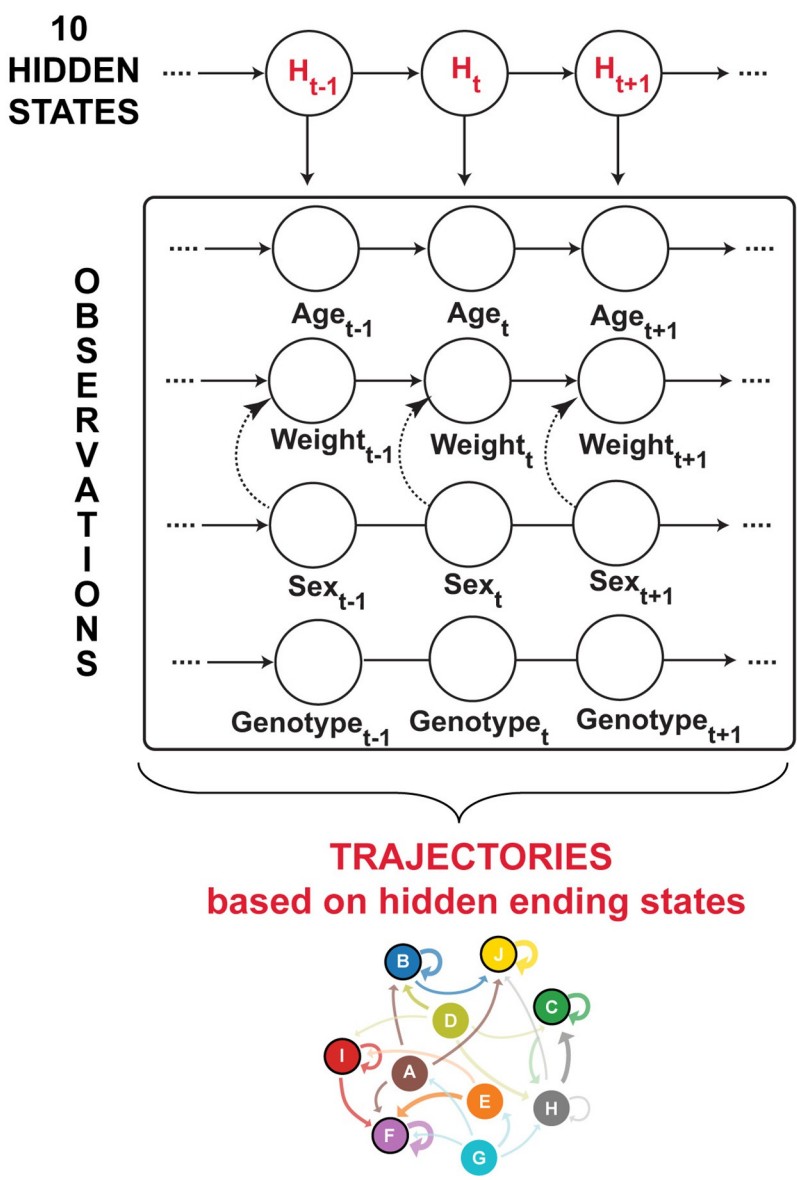

**Fig 2. Autoregressive Hidden Markov Model (AHMM) structure and output.** The schema illustrates the AHMM structure, highlighting the hidden states (H) learned by the model based on the real data observations, including age, weight, sex, and APOE genotype of our aging humanized APOE mouse colony. To account for the observed sex-dependent weight differences, where male mice weight more than female (Fig 1C), an interaction term was introduced between the variables sex and weight (dashed arrow). Based on these data, AHMM learns the transition probabilities between the hidden states enabling the identification of main weight trajectories. Relevant weight trajectories are selected by grouping together mice ending in same hidden states, allowing the study of the identified subgroups.

between longitudinal weight measurements and aging. Moreover, AHMM can be used in for time series of different lengths with missing data points or not equally spaced in time—an advantage of AHMMs over standard HMMs. Note that the model does not impute missing data or generate new data.

Observations in AHMM can be either continuous or discrete. In this study, hAPOE mice data included longitudinal weight measurements, along with age, sex, and APOE genotype.

Therefore, we adapted the AHMM to a mix of continuous (age and weights) and discrete (sex and genotype) variables.

Furthermore, the AHMM structure was adjusted to account for observed sex-based weight differences, as male mice weights on average are higher than female weights (Fig 1C). To account for this sex-weight dependency, an interaction term was introduced between the variable sex and the variable weight to account for their dependence (Fig 2). This adjustment aims to improve the model's convergence and accounts for the biological relation between sex and weight.

## Weight trajectory identification based on AHMM

The proposed AHMM learns from real weigh data, uncovering temporal patterns within longitudinal weight data based on the transition probability matrix A. In this study, as we are considering 10 possible hidden states, the resulting A matrix is a 10 by 10 matrix (Fig 3A) where each matrix element corresponds to the probability of transitioning between one state to another. Matrix A enables the identification of starting, intermediate, and ending states. Starting states are characterized by low incoming probabilities, while intermediate states exhibit non-zero in and out probabilities. Ending states are identified with high probabilities on the diagonal $P(A(a_{i,j}))$.

In our approach, mice that end in the same hidden state are considered to follow a unique weight trajectory and are grouped accordingly. Specifically, relevant weight trajectories are identified by grouping mice that end in the same hidden state and with a probability greater than 0.3 $(P(A(a_{i,j})))>0.3$.

AHMM was implemented using the Matlab Bayes Net Toolbox (BNT) adopted to work with both categorical (sex and genotype) and continuous observations (age and weight). In

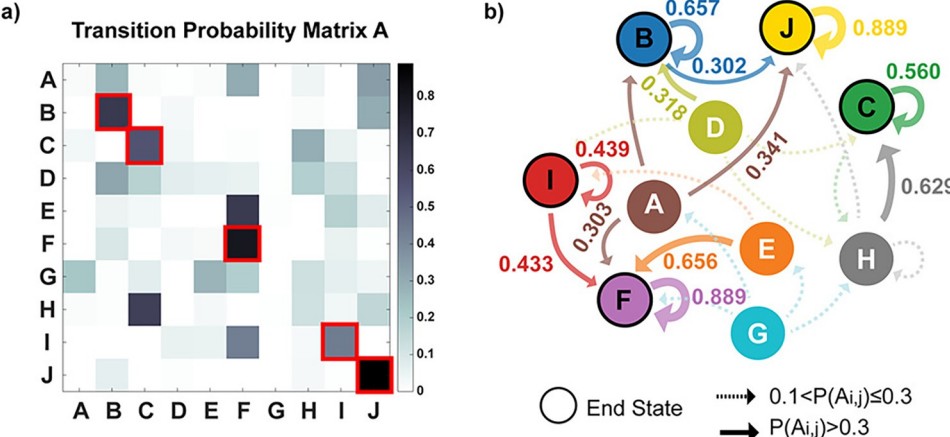

**Fig 3. AHMM transition probability matrix and diagram. a)** The matrix A illustrates transition probabilities between the possible 10 hidden states, with key ending hidden states (B,C,F,I,J) highlighted in red. Every hidden state is included as a row and as a column, and each cell in the matrix refers to the probability of transitioning from its row's state to its column's state. Numerical values of matrix A are provided in S2 Table. Key ending state are defined by a probability of arriving and staying in that state greater than 0.3 $(P(A(a_{i,j})))>0.3$. Mice ending in the same hidden states are part of a trajectory and are therefore grouped together. **b)** Diagram of the Markov Chain related to the transition matrix A. Nodes correspond to hidden states, and edges illustrate the transitions between them (changes of states). Nodes with black circles indicate the five most probable ending states. Arrow thickness indicates transition probabilities, with thicker arrows associated with higher probabilities of transitioning from or remaining in a hidden state. The diagram depicts starting, intermediate, and ending states, where starting states have more exiting arrows, and ending states are identified by loop arrows. In detail, key ending states (B,C,I,F, and J) are characterized by thicker loops, indicating higher probabilities of staying in those states when reached.

this study, we used the EM algorithm [34] to infer the parameters of AHMMs from data of our cohort. To ensure stability of the results, we repeated the EM algorithm 10 times and the AHMM with the largest resulting likelihood value was selected as the main model. Each run lasted 1,500 iterations to ensure convergence. Inference was performed on the resulting models using the junction tree algorithm [35].

## Weight trajectory validation

Weight trajectories identified by the AHMM were validated using post-hoc survival analysis and independent measurements of plasma β-amyloid (Aβ) concentration and EchoMRI observations of body composition.

**Survival analysis.** Survival analyses were performed considering the time from date of birth to death or to date of last weight measurement. Kaplan-Meier survival curve were computed R package *survival* [36]. Log-rank test and related p-value were computed to test between-group significance using *survdiff* R function [36]. Since our mouse cohort is used for a variety of experiments and research projects, we excluded from survival analysis mice that were sacrificed for experiments or other specific reasons.

## Plasma Aβ

Post-hoc analysis of plasma Aβ levels was conducted on plasma samples of mice resulting in distinct weight trajectories. Plasma samples to be included for Aβ analysis had to meet the following criteria: 1) collected between 12 to 14 months of age (when plasma Aβ is reliably detectable), and 2) balanced for sex and APOE genotype.

Animals were anesthetized using isoflurane and the diaphragm was severed. Whole blood was collected through cardiac puncture, deposited in Vacutainer tubes containing 3.6mg EDTA and placed on ice. Blood samples were then spun at 2,700 rpm for 7 min at 4˚C to separate the whole blood into its components. 100μL aliquots of plasma were taken and stored at -80˚C. Animals were subsequently perfused using ice-cold sterile PBS and tissues were collected. Collected tissues were immediately flash-frozen in liquid nitrogen before being stored at -80˚C.

Plasma samples for Meso Scale Discovery (MSD) use were selected by availability and trajectory from the available tissue bank. For plasma Aβ peptide concentration readings, the Mesoscale Discovery V-PLEX Plus Aβ Peptide Panel (4G8) Kit was used (a highly sensitive sandwich immunoassay), which allowed for rapid measurement of levels of the target protein in small samples. Plasma samples were pulled from -80˚C and allowed to thaw on ice. Samples were then diluted 1:4 using manufacturer supplied diluent. Calibrator and control solutions were prepared as per the manufacturer's instructions. Plates were blocked for 1 hour with 750 rpm shaking, washed, and detection antibody, controls, samples, and calibrators were added before shaking for an additional 2 hours. Plates were washed and immediately read via MESO QuickPlex SQ 120MM and analyzed using MSD DISCOVERY WORKBENCH analysis software.

## EchoMRI

Post-hoc analysis of EchoMRI was conducted on mice belonging to the resulting weight trajectories that were alive and not used for other experiments. Furthermore, data included for this analysis had to be collected from the same mice whose plasma Aβ levels were available.

EchoMRI (EchoMRI™ 3-in-1; Echo Medical System, Houston, TX) allows for the awake and unanesthetized measurement of body fat and lean mass. Equipment calibration occurred daily with a canola oil system test sample (COSTS). Animals were placed into specimen

holders, movement limited by a cylindrical insert, and placed into the antenna. A total of 9 acquisition scans were acquired and averaged per animal. The adipose index (percent body fat) was calculated using: [Fat mass / (Fat mass + Lean mass + Free water)] * 100.

### Supplementary validation using cross-sectional data

Supplementary validation of the trajectories identified by the AHMM was conducted using available Cross-Sectional (CS) data from mice included in the trajectories. The CS data consisted of single data points per mouse and comprised measurements of fasting blood glucose, CatWalk™ XT, EchoMRI™, and Novel Object Recognition (NOR). These data were not uniformly available across all mice and collected at various ages (S1 Fig) from different mice within the hAPOE colony. Due to these limitations, we adopted an alternative validation approach. For each mouse, we utilized sex, APOE genotype, weight measurement, and age at which the CS data was collected as inputs for the trained AHMM, allowing to subsequently infer hidden state for that time point. To assess statistical differences, we compared the distributions of the CS data based on the inferred hidden state (S2 File). CS data were collected following the procedures presented in S1 File, and the adopted validation methodology is described in S2 File.

## Results

### Weight trajectories derived from longitudinal data

The study cohort included 1,196 mice carrying hAPOE alleles, specifically hAPOE3/3, hAPOE3/4, or hAPOE4/4. The majority of mice were consistently weighed more than 3 or 4 times totaling 11,245 datapoints. In detail, 814 (83%) mice were weighed more than 5 times, and 494 (50%) were weighed more than 10 times (Fig 1A) over their lifespan. The cohort was balanced for sex, with 45% female and 55% male mice, and the APOE genotype distribution was 30% hAPOE3, 28% hAPOE3/4, and 42% hAPOE4/4 (Fig 1A). The weight data collected along with sex and APOE genotype information are reported in S1 Table.

Longitudinal weight data of hAPOE mice along with their age, sex, APOE genotype were used as input variables the AHMM (Fig 2) which identified the hidden states and their transition between them governing the input observations. 10 possible hidden states (A, B, C, D, E, F, G, H, I, and J) were considered in the AHMM (Fig 2) and mice ending in the same hidden state are considered part of a trajectory and are therefore grouped together. The AHMM transition matrix A resulting from this study is presented in Fig 3A, while the related transition diagram in Fig 3B.

Five most probable ending hidden states ($P(A(a_{i,j}))>0.3$) were identified based on the AHMM transition probability matrix A (Fig 3A) that corresponded to the states B, C, I, F and J (red squares in Fig 3A and loop arrows in Fig 3B). These five states correspond to stable ending states, while the remaining states (A,D,E,G, and H) did not result as ending stable states, corresponding to starting or intermediate states.

Mice were grouped according to the five stable ending states and the percent of weight difference for the mice following the same trajectory was computed between 12 months of age to the end of each trajectory. These resulted in -18%, -15%, 46%, -20%, and 5%, for the trajectories B, C, I, F and J, respectively, where negative values correspond to losing weight, while positive values to gaining weight (Fig 4J). Note that, not all the mice can be classified in one of the stable end trajectories, 17% (215) of mice in our colony did not have a stable end state and were not further explored. These mice correspond to outliers or mice assigned to intermediate or initial unstable states (S2 Fig).

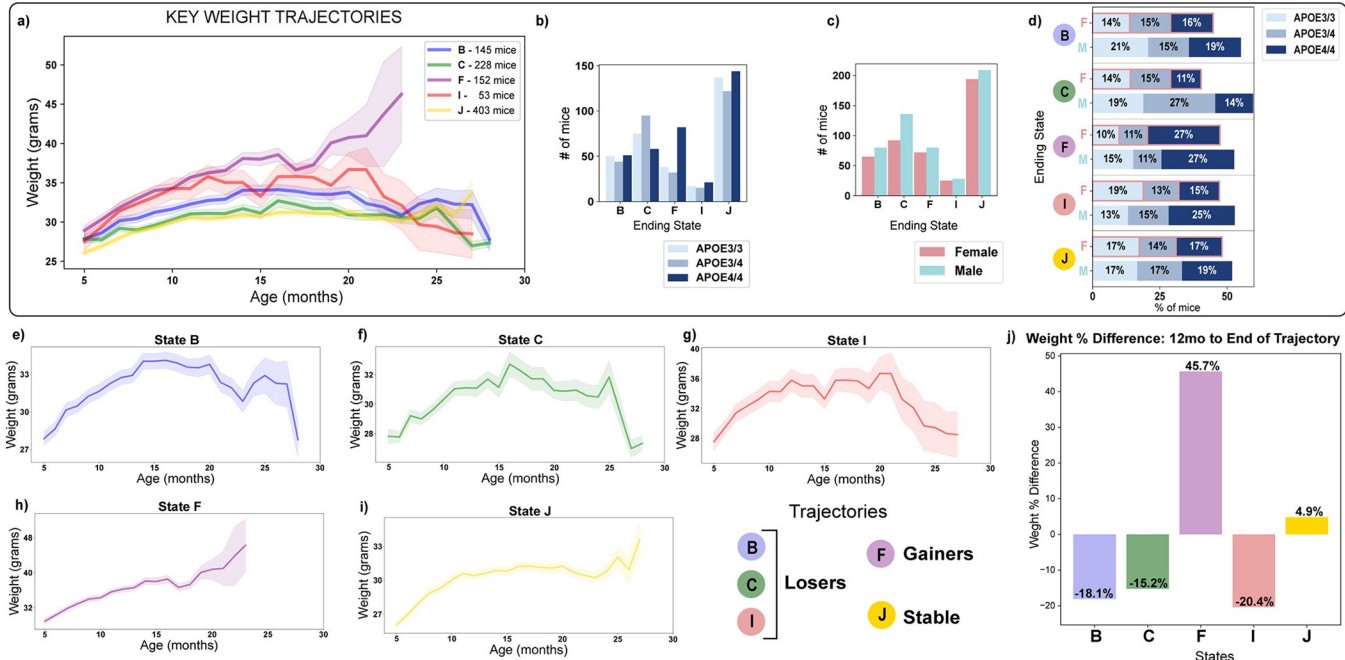

**Fig 4. Key weight trajectories in hAPOE mouse colony.** Panel **a)** The five most probable weight trajectories (labeled according to their hidden ending states) were derived using Autoregressive Hidden Markov Model (AHMM). The legend includes the number of mice within each trajectory. Panels **b-c)** Distribution of sex and APOE genotype within each identified weight trajectory. Panel **d)** Percent of mice by sex and genotype for each trajectory. Panels **e-i)** Visualization of each weight trajectory with separately fitted y-axes to enhance the clarity of the trend for each trajectory. The trajectories were obtained by plotting the mean and standard deviation for each age point meeting the minimum requirement of 3 mice per age. Panel **j)** presents the percent difference of weights for each trajectory from 12 months of age to the end of each trajectory. We selected 12 months to calculate weight percent differences as mice are in a growth phase until about 12 months (Figs 1B and 4A) and mice of 10–12 months are considered middle age [37].

Based on these results, among the five trajectories, the three trajectories B, C, and I exhibited weight loss trajectories for 426 (36%) mice (Fig 4E–4G). Trajectory F was composed of overall heavier mice and associated to weight increase in 152 (13%) mice (Fig 4H). Group J exhibited less than a 5% weight change (Fig 4J) consisting of 403 (34%) mice that exhibited a stable weight with no gain or loss (Fig 4I). Distributions of sex and APOE genotype for the identified trajectories are reported in Fig 4B–4D, and the relative trajectories by sex and APOE genotype are reported in S3 Fig.

The distribution of APOE genotypes were significantly different across the weight trajectories (chi-square test, p-value<0.05). Trajectory F exhibited the highest percent of APOE4/4 carriers (54%) exhibiting a significant difference in the frequency of E4 carriers compared to non-E4 carriers (chi-square test, p-value<0.05). In contrast, trajectory C was characterized by a higher proportion of males (60%) and APOE3/4 genotype (42%).

While sex did not vary significantly across all different trajectories (chi-square test, p-value>0.05), pairwise comparison across trajectories sex distribution revealed a significant difference between C and J (chi-square test, p-value<0.05).

## Key weight trajectory validation with survival analysis, Aβ and EchoMRI data

To validate the model and evaluate the identified trajectories, post-hoc survival analyses were conducted to assess differences in the survival rates across five identified phenotypes (Fig 5). After excluding mice that were sacrificed for experimental purposes or other reasons,

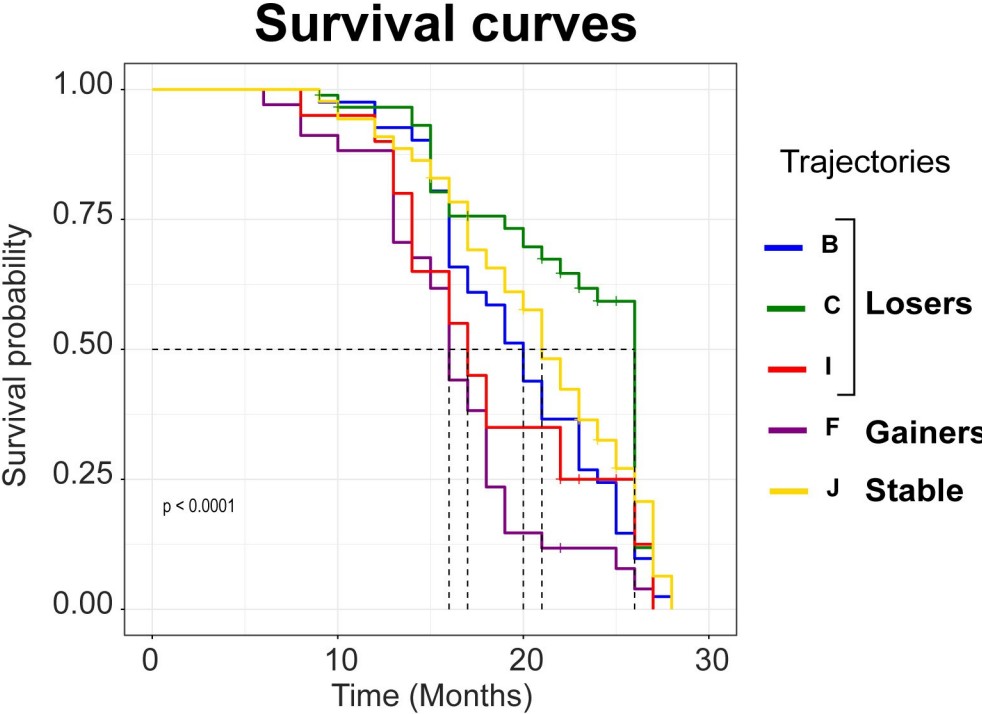

**Fig 5. Survival curves across weight phenotypes.** Survival curves associated with each of the five weight trajectories identified (B, C, I, F, and J). Significant difference in overall survival was found (log-rank p-value = 4e-09). For each state, median survival is pointed out with dash line corresponding to a survival probability of 0.5.

naturalistic survival data was available for 272 mice. The statistics on sex and APOE genotype distributions of these mice are reported in S4 Fig.

Survival data analyses revealed a significant difference in overall survival based on the trajectory group (Fig 4, log-rank p-value = 4e-09). Mice in trajectory F exhibited the poorest survival curves, followed by mice in trajectory I, B, J, and C (Fig 5).

Further, to evaluate whether the AHMM model could effectively identify distinct groups of mice based on longitudinal weight data and if the identified trajectories were also concurrently exhibiting distinctly different plasma Aβ levels, plasma samples were selected for a total of 49 mice ensuring balanced representation of various sexes, APOE genotypes, and weight change (S4C and S4D Fig). To mitigate the potential influence of age on plasma Aβ concentration, plasma samples from mice aged 12 to 14 were included in the analysis. For the C trajectory, n = 18 with 7 females and 11 males at an average age of 14 months (Standard Deviation (SD) = 1.36). For the F trajectory, n = 14 with 5 females and 11 males at an average age of 14.95 months (SD = 1.57). For the J trajectory, n = 17 with 9 females and 8 males at an average age of 13.34 months (SD = 1.15).

Statistical analysis indicated significant differences in the average levels of Aβ40 and Aβ42 in plasma between the weight gain phenotype and both the weight loss and stable phenotypes (t-test, pvalue = 3.63e-03 for Aβ40 and p-value = 2.25e-04 for Aβ42), as well as between the gaining phenotype and the stable phenotype (t-test, p-value = 5.12e-05 for Aβ40 and p-value = 9.54e-07 for Aβ42) (Fig 6A and 6B). Significant differences by sex were also found in trajectory C and F for average Aβ42 levels (t-test, p-value = 3.37e-02 for state C and p-value = 2.00e-02 for state F) (Fig 6C and 6D). Analysis of differences across genotypes were

## Plasma Aβ

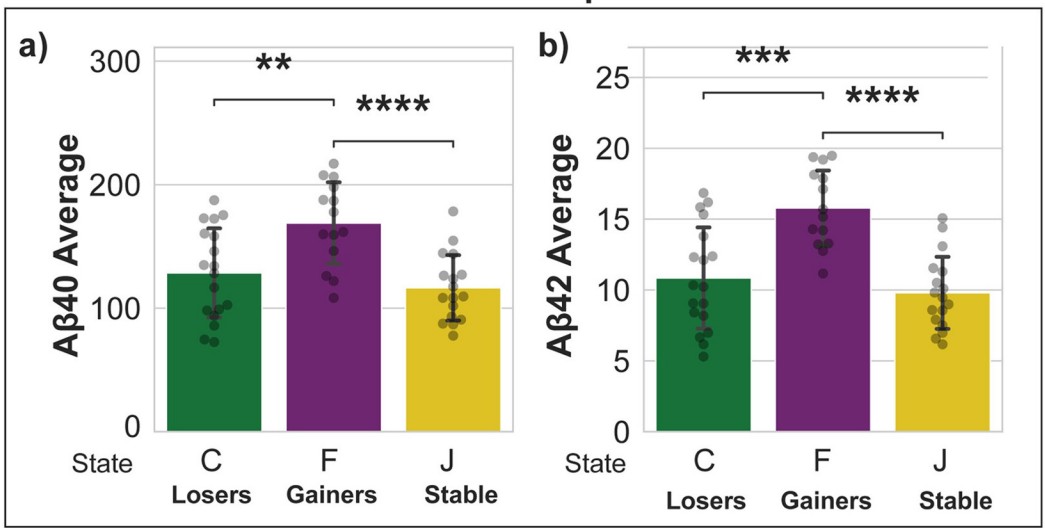

## Plasma Aβ by Sex

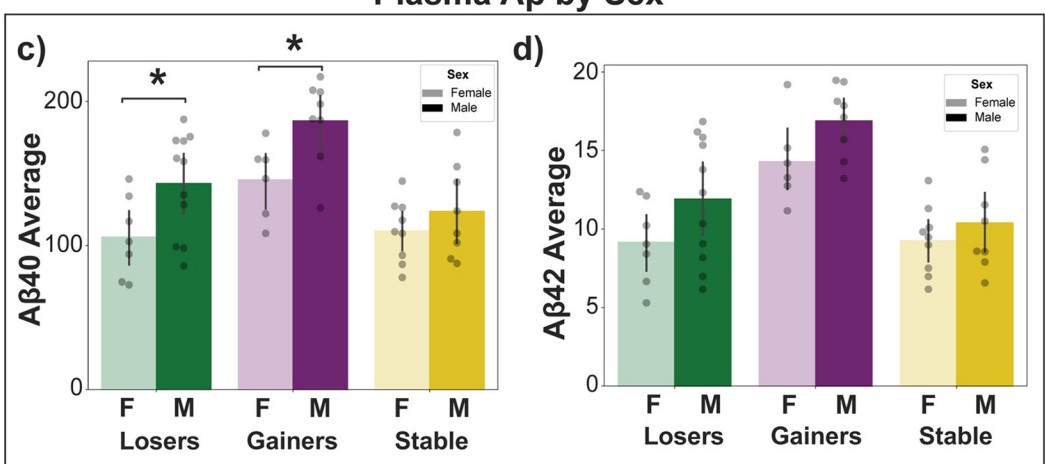

## EchoMRI

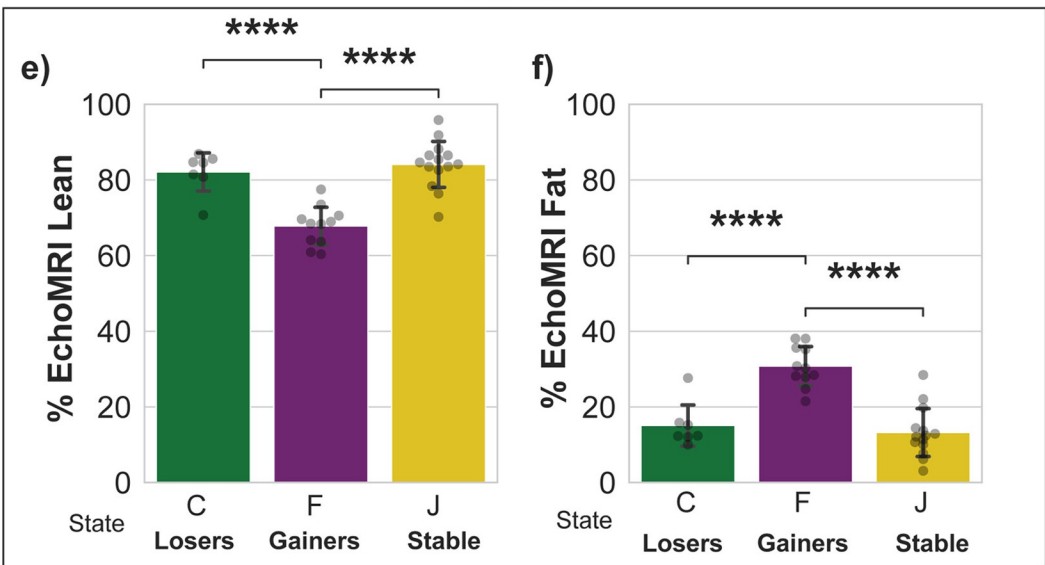

**Fig 6. Distributions of plasma Aβ concentrations and EchoMRI data across weight loss, gain, and stable trajectories.**
Panel **a-b** show respectively average level of plasma Aβ40 and Aβ42 concentrations. Panel **c-d** shows average level of plasma Aβ40 and Aβ42 concentrations by sex. Panel **e-f** shows the distribution of EchoMRI body compositions. *: 1.00e-02 < p-value < = 5.00e-02; **: 1.00e-03 < p-value < = 1.00e-02; ***: 1.00e-04 < p-value < = 1.00e-03; ****: p-value < = 1.00e-04.

not powered enough and the distribution of plasma Aβ40 and Aβ42 concentrations by APOE genotypes are reported in S5 Fig.

EchoMRI™ data were available for 32 mice of the 49 included in the plasma Aβ analysis, with 7 belonging to the losing phenotype, 11 to the gaining phenotype, and 14 to the stable phenotype. The statistics on sex and APOE genotype distributions of these mice are reported in S4E and S4F Fig, respectively.

Statistical analysis of EchoMRI™ data across the weight trajectories indicated significant differences in body composition between the gaining and the losing phenotypes (t-test, fat percentage p-value = 4.44e-05 and lean percentage p-value = 2.65e-05) as well as between the gaining and the stable phenotypes (t-test, fat percentage p-value = 4.81e-07 and lean percentage p-value = 2.70e-07) (Fig 6E and 6F).

## Supplementary validation results on cross-sectional data

Supplementary validation of the trajectories identified by the AHMM conducted using additional Cross-Sectional (CS) data: (i) 132 fasting blood glucose, (ii) 131 CatWalk™ XT, (iii) 298 EchoMRI™, and (iv) 133 Novel Object Recognition (NOR) observations (S3 Table).

The distributions of all four CS datasets exhibited significant differences across the hidden states inferred by the AHMM at the time of CS data collection (S6 Fig). Note that for this analysis differently from the one presented in Fig 6, CS data were not uniformly collected across all mice and collected at the various months of age (S1 Fig, S3 Table). For this reason, these results are presented in the supplementary material to support the validation of the AHMM (S6 and S7 Figs, S3 Table).

## Discussion

The prodromal phase of AD can occur 10–20 years prior to AD diagnosis [38]. Identification of biological indicators associated with the prodromal phase of AD is critical for prevention or treatment in the early stages of AD, to identify patients at higher risk and delay the onset and progression of the disease. Weight changes have been observed in humans prior to AD diagnosis, suggesting their potential as preclinical markers of AD [13,16,39]. Determining weight changes that occur in AD mouse models, such as the humanized APOE transgenic mouse model, could provide valuable insights and potential therapeutic targets [40–46].

In late onset Alzheimer's, the greatest non-modifiable risk factors are age, chromosomal sex, and APOE4 genotype. These major drivers are incorporated into our hAPOE mouse model colony establishing it as a valid model for studying the functional roles of human APOE genotypes in AD-related processes. Due to the homogeneity of dietary and environmental conditions, it is often assumed that mice age uniformly. However, this study revealed a hidden complexity of biological aging through longitudinal measurements of weight based on age, APOE genotype, and chromosomal sex. By utilizing these variables, we uncovered the distinct dynamics of aging, which are critical for the translational validity of AD mouse models. As in the human population, outcomes of this study demonstrate that there are distinct trajectories of resilience and trajectories of vulnerability.

Results reported herein are based on a comprehensive analysis of longitudinal weights from a colony of 1,196 aging hAPOE mice exposed to the same diet, totaling 11,245 datapoints

collected from November 2nd, 2018, to January 28th, 2022. The application of the advanced AHMM algorithm enabled the identification of five weight trajectories, encompassing patterns akin to those observed in human studies, including weight gain, loss, and stability. Specifically, trajectories B, C, and I (36% of mice) were associated with weight loss, trajectory F (13% of mice) exhibited weight gain, and trajectory J (34% of mice) represented a stable state with no significant weight changes. These trajectories, undetectable through simple averaging (Fig 1B and 1C), underscore the importance of longitudinal, colony-wide assessments in revealing heterogeneity resembling human phenotypes.

Notably, weight gain trajectory F represented a unique subset of heavier mice with lower survival curves and significantly higher plasma Aβ concentrations when compared to the weight loss and stable trajectories. The data indicated that trajectory F was unique in that these mice exhibited a 45.7% increase in body weight starting at midlife (12 months) for both females and males and was more prevalent in APOE4 carriers although not exclusive to APOE4 carriers. Further, trajectory F mice had significantly greater body fat composition when compared to trajectories of weight loss or weight stability. Translationally trajectory F is consistent with a recent study reporting that adiposity is associated with cognitive decline [20]. Collectively, these data indicate an association between weight gain, APOE4 carrier status, Aβ generation, and earlier death [16,47,48]. Obesity and diabetes are known risk factors for AD in humans, and this subgroup of mice may resemble a subpopulation prone to these conditions [47,49–51].

Among the three weight loss trajectories (B, C, I), trajectory B was characterized by a sharp drop in weight at 23 months of age, occurring later in life in comparison to trajectories C and I. Trajectories C and I exhibited earlier weight loss at 15 months of age, roughly equivalent to the human age of 50 years, which aligns with prodromal weight decline prior to AD diagnosis [12,21,39]. Although these two trajectories appear similar, trajectory C started with lower body weight and was associated with better survival curves compared to trajectory I which was characterized by mice that were initially heavier and who exhibited lower survival rates indicating a potentially worse prognosis for heavier mice.

The weight trajectory J captured a subgroup of mice with stable weights and had greater survival relative to trajectories B, F, and I which could represent a resilient human population or with a better AD prognosis.

The distribution of APOE genotypes across all weight trajectories was significantly different. Specifically, trajectory F exhibited the highest percent of hAPOE4/4 carriers (54%). Trajectory C, among the others, had a higher proportion of hAPOE3/4 carriers (42%). Furthermore, trajectory C exhibited a sex difference with 60% of the mice being male. Interestingly, stable trajectory J was consistent across APOE genotype.

Trajectory I was predominately characterized by hAPOE4/4 males (25% of mice in trajectory I) and hAPOE3/3 females (19% of mice in trajectory I). The weight loss patterns observed in trajectories C and I were consistent across both female and male mice. Weight loss was more evident for hAPOE3/4 and hAPOE4/4 carriers in trajectory C and for hAPOE3/3 and hAPOE4/4 carriers in trajectory I. The heterogeneity in weight loss observed in trajectories C and I aligns with the weight heterogeneity in human AD patients [52] and was characterized by lower initial weight and progressive weight decline. Surprisingly, trajectory C, despite exhibiting weight loss, was associated with longer survival, primarily driven by hAPOE3/4 mice.

Validation of AHMM trajectories was conducted utilizing independent survival, plasma Aβ and EchoMRI mouse data that were not included for generation of the AHMM model. These data confirmed differences in the identified trajectories. In detail, post-hoc survival analysis revealed statistically significant differences in overall survival across the five identified

trajectories. Trajectory F and I had the poorest survival rates, a higher proportion of hAPOE4/4 genotypes, and overall heavier mice. In contrast, trajectory C, associated with weight loss, exhibited better survival rates and a higher proportion of mice with hAPOE3/4 genotype and males. Trajectory J (stable weight) and B (another weight loss trajectory) had intermediate survival curves. These survival patterns confirm distinct weight dynamics and heterogeneous aging profiles. Plasma Aβ levels revealed that weight gain trajectory F had the highest plasma Aβ levels and was also associated with a higher proportion of hAPOE4/4 carriers. Trajectories C and J also had detectable plasma Aβ which were essentially equal in magnitude and significantly lower than trajectory F.

Collectively, longitudinal analysis of weight in an aging cohort of both female and male mice across different APOE genotypes revealed five weight trajectories consistent with the diversity of weight change in the human population during aging [13,16,39]. In addition, supplementary analysis of cross-sectional data including fasting blood glucose, CatWalk, EchoMRI, and NOR measurements confirmed statistically significant differences across subgroups of mice with distinct weight trajectories identified by the AHMM. The importance of these additional analyses is that regardless the different ages of collection of these data, the AHMM is able to infer trajectories to observations of mice with specific age, sex and APOE genotype that are also associated with significant differences in physiological outcomes like blood glucose levels and on system levels outcomes like cognition or motor function.

Our findings of different aging phenotypes associated with weight changes mirrored those reported for humans. Ukraintseva et al. [53] reported weight loss in women who developed AD as early as their forties, long before clinical diagnosis and it is consistent with the perimenopause transition in women as a key biological process that can unmask later life AD (3). The Ukraintseva et al. findings align with reports of inexplicable weight loss that can precede AD by decades and indicated that biological aging in APOE4 carriers reflect an underlying, prolonged prodromal phase of AD [53]. This human trajectory of weight loss resembles trajectories I and C in our mouse colony. Furthermore, Ukraintseva et al. reported that association between AD and lower weight could not be explained solely by the effects of APOE4 [53], which is consistent with our findings of different weight trajectories of APOE4 mice.

Similarly, Holmes et al. [54] observed that APOE4 carriers were associated with lower survival rates and tended to reach peak weight at younger ages, followed by a steeper decline at later ages, a pattern consistent with mouse weight trajectories C and I.

In another study, Bell et al. [55] underlined the challenge of examining underlying mechanisms that explain the association between body mass index (BMI) and brain health to advance knowledge of AD. Bell et al. observed that higher late-life BMI was associated with a lower risk of incident dementia, however BMI is not protective in the presence of rapid weight loss [55]. Our findings in trajectory I, characterized by heavier mice with a rapid weight loss support this human weight phenotype. On the other hand, Bell et al. also highlighted that higher mid-life BMI is associated with increased AD risk which could be driven "by a long-standing burden of vascular and metabolic risk factors" [55]. Our findings within trajectory F support this pattern as trajectory F is characterized by sustained weight gain and overall heavier body weights throughout aging. Interestingly, trajectory F also displayed the poorest survival outcomes and a preponderance of APOE4 mice in our colony. This pattern parallels Bell et al.'s findings in human populations, where higher mid-life BMI correlates with increased AD risk and poorer health outcomes [55].

Overall, applying AHMM revealed complexity of aging within a mouse colony exposed to the same dietary and environmental conditions that would have otherwise remain hidden. Furthermore, this study demonstrated that the hAPOE mouse model successfully replicates the natural variability in weight change across aging mirrors human trajectories of weight

gain, loss, and stability. Importantly, these distinct trajectories were uniquely identified through the application of the AHMM model to our longitudinal data. One strength of this data was the extensive weight data collected longitudinally for 1,196 mice. A shortcoming of this dataset arises from the lack of survival and cross-sectional data for all the mice in the cohort. Ideally, to determine differences across trajectories at end hidden states, survival and CS data should have been collected for all the mice cohort and at the end of their life. However, our cohort is an aging cohort and are sacrificed for experiments and research. Thus, it was not possible to collect new cross-sectional data for most of the mice due to their lifespan and the planning for additional experiments is not possible. Furthermore, future studies will aim to expand the analysis to a larger colony, allowing for more comprehensive phenotyping of weight trajectories, with stratification based on intermediate states, in addition to their final trajectories. Future studies will extend this analysis to our currently aging colony of APP-A-POE mouse models, which incorporate the APP mutation associated with AD development and pathology. This will address the limitation of current hAPOE mouse models, which primarily represent AD risk. Furthermore, APOE2 mice were not included in our colony, the AHMM model provides a strategy to determine the aging trajectories in existing mouse colonies from different laboratories. Future directions will focus on investigating whether APOE2 genotype alters the aging trajectories or whether APOE2 mice exhibits the same variability in aging trajectories comparable to those we detected for APOE3/3, APOE3/4 and APOE4/4 mice.

Our findings show that a contributing factor to the variability in the translational validity of preclinical mouse analyses is the assumption that all mice will age similarly. Our data indicate that, despite standardized vivarium maintenance practices, mice age differently exhibiting weight gain, loss, and stability consistent with the heterogeneity of changes in human weight across aging. The heterogeneity in aging mouse weight trajectories is consistent with the association of both weight gain and loss and increased risk for AD diagnosis [16,18–21,56]. Thus, the conventional assumption of homogeneity of a mice colony is not supported by the heterogeneous profiles that emerged from a fundamental indicator of aging, change in body weight.

In conclusion, analyses reported herein identified gain, loss, and stable weight trajectories during aging of an hAPOE mouse colony. Comprehensive characterization of weight change phenotypes of an aging hAPOE mouse colony has the potential to increase translational validity to human AD. These insights hold the potential to contribute to the development of more effective interventions for AD, ultimately bringing us closer to the goal of effectively preventing and treating this devastating disease.

## Supporting information

**S1 Table. Weight, age, sex, and APOE genotype of the mouse colony.**
(XLSX)

**S2 Table. Transition Probability Matrix with values.**
(XLSX)

**S3 Table. Cross-sectional data of the mice colony.**
(XLSX)

**S1 Fig. Distributions of age and weights at the time of collection of each cross-sectional data.** a) shows distribution of age for blood glucose, CatWalk, EchoMRI, and Novel Object Recognition (NOR), while b shows the distribution of mice weights in grams at the time of cross-sectional data collection. These distributions highlight the variability in age and weights of mice for which physiological and behavioral data were available. Cross-sectional data

available for a small subset of mice: (i) 132 fasting blood glucose, (ii) 131 CatWalk™ XT, (iii) 298 EchoMRI™, and (iv) 133 NOR observations. Cross-sectional data are reported in S3 Table. (TIF)

**S2 Fig. Weights over time for the mice assigned to the less probable and not stable states.** Lines are colored according to the last not stable hidden state (A, D, E, G, or H) associated to the mouse assigned by the Autoregressive Hidden Markov Model (AHMM). (TIF)

**S3 Fig. Key weight trajectories in hAPOE mouse colony by sex and APOE genotype.** a) shows the 5 key weight trajectories (B, C, I, F and J) derived using Autoregressive Hidden Markov Model (AHMM) split by sex. b) shows the AHMM identified trajectories by APOE genotype. Legends report the number of mice belonging to each curve (subgroup). (TIF)

**S4 Fig. Survival, Aβ plasma, and EchoMRI Data overview.** Percent of sex and APOE genotype distributions of survival data (panels a and b), Aβ plasma data (panels c and d), and EchoMRI™data (panels e and f). Panel a, c, and e show survival, Aβ plasma, and EchoMRI™ based on sex. Panels b, d, and f e show survival, Aβ plasma, and EchoMRI™ based on genotype. N indicates the number of mice in each dataset. (TIF)

**S5 Fig. Distributions of Aβ concentration by APOE genotype across weight loss, gain, and stable trajectories.** Panels a and b show average levels of Aβ40 and Aβ42 concentrations, respectively. Bars featuring only the outline color represent groups with fewer than 3 samples. (TIF)

**S6 Fig. AHMM estimated hidden states are associated with differences in cross-sectional physiological and behavioral outcomes. a)** shows distribution of age for blood glucose, CatWalk, EchoMRI, and Novel Object Recognition (NOR), while b shows the distribution of mice weights in grams at the time of cross-sectional data collection. These distributions highlight the variability in age and weights of mice for which physiological and behavioral data were available. Panels **a)** to **d)** show blood glucose, CatWalk, EchoMRI, and NOR across AHMM-inferred hidden states. Statistical differences between data distributions across the estimated trajectories are highlighted with an asterisk. P-value annotation legend: *: $1.00e-02 <$ p-value $< = 5.00e-02$; **: $1.00e-03 <$ p-value $< = 1.00e-02$; ***: $1.00e-04 <$ p-value $< = 1.00e-03$; ****: p-value $< = 1.00e-04$. The AHMM learned transition probabilities from the longitudinal weight data was utilized to estimate the hidden state (potential trajectory group) by including the same variables used as input (age, sex, APOE genotype, and weight) for the cross-sectional data. Using the AHMM probabilities, each CS measurement was associated with one of the 10 possible hidden states. Glucose blood concentration, EchoMRI, CatWalk, and NOR values were subsequently grouped according to the inferred hidden ending state. To determine if significant differences existed between hidden states, we conducted t-tests on the data distributions. (TIF)

**S7 Fig. Age and weight boxplots of Cross-Sectional (CS) data for the inferred hidden states.** Panel a-j show for each inferred hidden state distribution of the age at the data collection, weight at the data collection, and distribution of the data respectively for EchoMRI, CatWalk, Glucose, and NOR. In the boxplots, straight lines denote that mice in the respective trajectories share the same age, while dots represent outliers. (TIF)

**S1 File. Cross-Sectional data collection method.**
(DOCX)

**S2 File. Autoregressive Hidden Markov Models trajectory validation.**
(DOCX)

## Author Contributions

**Conceptualization:** Francesca Vitali, Allan Tucker, Roberta Diaz Brinton.

**Data curation:** Francesca Vitali, Jean-Paul Wiegand, Lillian Parker-Halstead.

**Formal analysis:** Francesca Vitali.

**Funding acquisition:** Roberta Diaz Brinton.

**Investigation:** Francesca Vitali.

**Methodology:** Francesca Vitali, Jean-Paul Wiegand, Allan Tucker.

**Project administration:** Francesca Vitali, Jean-Paul Wiegand.

**Resources:** Francesca Vitali.

**Software:** Francesca Vitali.

**Supervision:** Francesca Vitali.

**Validation:** Francesca Vitali.

**Visualization:** Francesca Vitali.

**Writing – original draft:** Francesca Vitali, Jean-Paul Wiegand, Allan Tucker, Roberta Diaz Brinton.

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
