## [Decision Letter · Decision Letter 0]

2 Sep 2024

PONE-D-24-34922Weight Trajectories in Aging Humanized APOE Mice with Translational Validity to Human Alzheimer's Risk PopulationPLOS ONE

Dear Dr. Vitali,

Thank you for submitting your manuscript to PLOS ONE. After careful consideration, we feel that it has merit but does not fully meet PLOS ONE’s publication criteria as it currently stands. Therefore, we invite you to submit a revised version of the manuscript that addresses the points raised during the review process.

We look forward to receiving your revised manuscript.

Kind regards,

Maud Gratuze

Academic Editor

PLOS ONE

Journal Requirements:

"Research reported herein was supported by the National Institute on Aging (grants P01AG026572 [Perimenopause in Brain Aging and Alzheimer's Disease], T32AG061897 [Translational Research in Alzheimer's Disease and Related Dementias (TRADD)], 5R01AG057931-02 [Sex Differences in Molecular Dementias of Alzheimer's Disease Risk: Prodromal Endophenotype]), the Women's Alzheimer's Movement to Roberta Diaz Brinton, and the University of Arizona Center for Innovation in Brain Science."

3. In this instance it seems there may be acceptable restrictions in place that prevent the public sharing of your minimal data. However, in line with our goal of ensuring long-term data availability to all interested researchers, PLOS’ Data Policy states that authors cannot be the sole named individuals responsible for ensuring data access (http://journals.plos.org/plosone/s/data-availability#loc-acceptable-data-sharing-methods).

Reviewers' comments:

Reviewer's Responses to Questions

**Comments to the Author**

1. Is the manuscript technically sound, and do the data support the conclusions?

Reviewer #1: Partly

Reviewer #2: Yes

2. Has the statistical analysis been performed appropriately and rigorously? 

Reviewer #1: I Don't Know

Reviewer #2: Yes

3. Have the authors made all data underlying the findings in their manuscript fully available?

Reviewer #1: Yes

Reviewer #2: Yes

4. Is the manuscript presented in an intelligible fashion and written in standard English?

Reviewer #1: Yes

Reviewer #2: Yes

5. Review Comments to the Author

Reviewer #1: The real impact of the paper is limited to understanding how mouse models can vary over time, rather than representing real meaningful data for AD diagnosis or therapeutics. The results do offer some insight into variability within mouse models despite equal conditions, but extrapolating that to AD is difficult. It is crucial to distinguish between noatural biolgical aging and AD pathology, and both are accompanied by weight loss following metabolic changes over time. The authors use a model with humanized APOE, however the model is not an AD mouse model, therefore correlating the findings specifcally to the pathology is not entirely reasonable. The authors should reword parts of the manuscript to specify these limitations. The scope of the paper should therefore be limited to enhancing the reproducibility and universality of mouse models, rather than hinting towards potential future therapeutic targets for AD based on the results obtained.

Furthermore, the authors use APOE3 and APOE4 animals, which is relevant given the risk associated to APOE4. However, the lack of APOE2 in the study is disappointing, given the protective role of APOE2 in AD, and its links to hyperlipoproteinemia. Expaning the study to all APOE genotypes, and using an AD mouse model (for example the 5XFAD model) would greatly enhance the importance of these results.

Aside from these general questions, I have some remarks regarding the manuscript:

1) In the abstract, higher body fat is associated to APOE4 carriers, whereas APOE3/4 carriers (which are still APOE4 carriers) are linked to weight loss. This discrepancy is highly confusing and should be made much clearer.

2) The authors mention that animals were weighed once a month, however the inclusion criteria is of animals of at least 5 months that were weighed at least 3 times. In the results section, they expand by saying 50% of animals were weighed over 10 times, whereas 17% were wieghed less than 5. This indicates that, in the case of the older mice particularly (which are also the most relevant mice for the study), vast amounts of data are likely generated through the probabilistic model. Although the AHMM allows the prediction of missing values, I question the validity of such large amounts of data that are generated through a model. I was expect at the very least for half of the data points to be true data, which does not appear to be the case. The authors should reconsider how they present this data, and perhaps offer more insight into the specific number of times each age group was weighed, and with how things are represented it's possible that some 28-month-old mice weighed only weighed a few times, which is not trustworthy.

3) There is no section 2.4.4 int he methods.

4) The % of each APOE genotype varies slightly between Fig 1A and the results section.

5) In Line 277, the red squares should be referenced to figure 3A rather than 2A.

6) In lines 408-409 the total sum of percentages is 83%, leading to doubts regarding the missing 17%. As the first part of the discussion, the authors should mention what this missing % is referring to.

Reviewer #2: This study reports of weight changes that occur in the humanized APOE transgenic mouse models, ones oftentimes used to model AD due to APOE4 being the strongest risk factor for AD. The authors used Autoregressive Hidden Markov Model (AHMM) to predict weight trajectories of the hAPOE ageing mice of both sexes, demonstrating that there are distinct weight, resilience and vulnerability trajectories in these mice models as there are in human population. Even though the study is statistically sound and well rounded, I suggest the following points to be re-evaluated:

1. The study is based on the effect of the APOE risk factor on the weight trajectories of the ageing mice, however not much if anything has been directly mentioned about the weight trajectories of human APOE4 carriers. This has been previously reported by Ukraintseva, et al., 2024; Holmes et al., 2024; Bell et al., 2017; Backman et al., 2015 and others. I would therefore suggest elaborating more on weight trajectories in human APOE4 carriers and linking them with the results obtained in this study.

2. As a continuation of the 1st suggestion, it has been shown by Holmes and colleagues (2024) that "APOE4 carriers have 19%–22% (TE p = 0.020–0.039) lower chances of surviving to age 85 and beyond, in part, because they reach peak values of weight at younger ages, and their weight declines faster afterward compared to non-carriers."

Such weight trajectory would correspond to trajectories C and I detected in this study, as authors also mentioned the link between these trajectories and prodromal weight decline prior to AD diagnosis (line 429). However, the percentage of APOE4 carriers (male and female) was highest in the F trajectory. Thus, it would be informative to elaborate on these results in more detail.

6. PLOS authors have the option to publish the peer review history of their article (what does this mean?). If published, this will include your full peer review and any attached files.

Reviewer #1: No

Reviewer #2: No

---

## [Author Response · Author response to Decision Letter 0]

18 Oct 2024

We thank the editor and the reviewers for their useful comments and suggestions. 

We addressed the requirements requested by the editor and: 1) adapted the manuscript to the PLOS ONE style template, 2) added the funders role to the financial disclosure, 3) included non-author institutional point of contact for data request, 4) added the title and caption of Supplementary Information files at the end of the manuscript after the References, 3) Reviewed manuscript references. 

Reviewer’s comments and concerns have been addressed as reported below and detailed in the Response to Reviewers file submitted to the journal. 

Reviewer #1: 

“The real impact of the paper is limited to understanding how mouse models can vary over time, rather than representing real meaningful data for AD diagnosis or therapeutics. The results do offer some insight into variability within mouse models despite equal conditions, but extrapolating that to AD is difficult. It is crucial to distinguish between natural biological aging and AD pathology, and both are accompanied by weight loss following metabolic changes over time. The authors use a model with humanized APOE, however the model is not an AD mouse model, therefore correlating the findings specifically to the pathology is not entirely reasonable. 

The authors should reword parts of the manuscript to specify these limitations. The scope of the paper should therefore be limited to enhancing the reproducibility and universality of mouse models, rather than hinting towards potential future therapeutic targets for AD based on the results obtained.”

We appreciate the reviewer’s feedback and acknowledge the importance of using accurate mouse models of AD. We revised the entire manuscript to clarify that this study aims at characterizing an aging colony of hAPOE mice which reflects key drivers of AD risk, rather than focusing on modelling AD pathology. Specifically, this study focuses on three major drivers of late onset AD: 1) aging, 2) chromosomal sex, and 3) APOE genotype, where the APOE4 isoform is the greatest genetic risk factor for AD. 

While we understand that our study does not include transgenic mouse models of AD, a recent review on AD mouse models confirmed JAX hAPOE mouse models as established models to study the functional roles of human APOE genotypes in AD-related processes (1). The importance of taking into account the impact of biological variables such as age, sex and human APOE genotypes on disease processes has been well-established for translational validity of mouse models of AD risk (1). Transgenic mouse models of familial AD are relevant to 1-2% individuals with AD. We chose to focus on the multivariate nature of aging in the context of sex and APOE genotype biology which is relevant to the vast majority of individuals at risk for developing AD and therefore relevant to development of therapeutics to prevent, delay and treat late onset AD. 

Our study investigates weight trajectories as a hallmark of aging in AD risk populations, examining the translational validity of weight changes in hAPOE mouse models. Whitin 11,245 weight measurements from 1,196 mice, our analysis provides a detailed retrospective study of weight trajectories, with the aim of enhancing the translational validity of mouse models of AD risk.

We revised the entire manuscript (e.g. Introduction Lines 78-84, Discussion Lines 408-410) to highlight the focus of this study on the translational validity of AD-risk-mouse-models. The importance of analyses reported in our study is that the variability observed in the mouse model on fundamental biological variables models the variability in risk factor profiles relevant to human risk of AD. 

To highlight the retrospective nature of this study we changed the title to: “Weight Trajectories in Aging Humanized APOE Mice with Translational Validity to Human Alzheimer’s Risk Population: a retrospective study”

Introduction Lines 78-84: “hAPOE mouse models are established models to study the functional roles of human APOE genotypes in AD-related processes (26). The importance of considering the impact of biological variables such as age, sex and human APOE genotypes on disease processes has been well-established for translational validity of mouse models of AD risk (26). In this study, we focus on the multivariate nature of aging in the context of sex and APOE genotype biology which is relevant to the vast majority of individuals at risk for developing AD and therefore relevant to development of therapeutics to prevent, delay and treat late onset AD.”

Discussion Lines 408-410: “In late onset Alzheimer’s, the greatest non-modifiable risk factors are age, chromosomal sex, and APOE4 genotype. These major drivers are incorporated into our hAPOE mouse model colony establishing it as a valid model for studying the functional roles of human APOE genotypes in AD-related processes.”

“Furthermore, the authors use APOE3 and APOE4 animals, which is relevant given the risk associated to APOE4. However, the lack of APOE2 in the study is disappointing, given the protective role of APOE2 in AD, and its links to hyperlipoproteinemia. Expanding the study to all APOE genotypes, and using an AD mouse model (for example the 5XFAD model) would greatly enhance the importance of these results.”

We agree with the reviewer that including the APOE2 genotype, which has a protective role in AD, would have further enhanced our study. However, the current analysis is a retrospective study of an existing aged colony of mice that does not include the APOE2 mutation. In addition, the APOE2 isoform is relatively rare, with only 5% incidence (2). 

While the current study focuses on APOE3/3, APOE3/4 and APOE4/4 genotypes, we acknowledge the importance of future investigations that incorporate APOE2. 

While APOE2 mice were not included in our colony, the AHMM model provides a strategy to determine the aging trajectories in existing mouse colonies from different laboratories.

It is an exciting question whether the APOE2 genotype alters the aging trajectories or whether APOE2 mice exhibits the same variability in aging trajectories comparable to those we detected for APOE3/3, APOE3/4 and APOE4/4 mice. 

We agree that a mouse model that generates AD pathology is an important avenue to pursue. In this regard, we are actively expanding our mouse colony to include APP-APOE mice and are collecting comparable data in these mice to determine the impact of the humanized APP gene on aging profiles and development of AD pathology. While the 5XFAD mouse is an excellent model of amyloidosis as to date there is no evidence in the human population of the coexistence of the five mutations present in the 5XFAD model.

To address the reviewer concerns we have added the following sentence to the Discussion section of the manuscript:

Discussion Lines 527-534: ‘Future studies will extend this analysis to our currently aging colony of APP-APOE mouse models, which incorporate the APP mutation associated with AD development and pathology. This will address the limitation of current hAPOE mouse models, which primarily represent AD risk. Furthermore, APOE2 mice were not included in our colony, the AHMM model provides a strategy to determine the aging trajectories in existing mouse colonies from different laboratories. Future directions will focus on investigating whether APOE2 genotype alters the aging trajectories or whether APOE2 mice exhibits the same variability in aging trajectories comparable to those we detected for APOE3/3, APOE3/4 and APOE4/4 mice.’

“Aside from these general questions, I have some remarks regarding the manuscript:

1) In the abstract, higher body fat is associated to APOE4 carriers, whereas APOE3/4 carriers (which are still APOE4 carriers) are linked to weight loss. This discrepancy is highly confusing and should be made much clearer.”

We thank the reviewer for making this point, we have clarified in the abstract that we are referring to APOE4/4 carriers 

Abstract: Weight gain trajectory was associated with elevated plasma β-amyloid levels, higher body fat composition, lower survival rates and a greater proportion of APOE4/4 carriers.

2) The authors mention that animals were weighed once a month, however the inclusion criteria is of animals of at least 5 months that were weighed at least 3 times. In the results section, they expand by saying 50% of animals were weighed over 10 times, whereas 17% were weighted less than 5. This indicates that, in the case of the older mice particularly (which are also the most relevant mice for the study), vast amounts of data are likely generated through the probabilistic model. Although the AHMM allows the prediction of missing values, I question the validity of such large amounts of data that are generated through a model. 

I expect at the very least for half of the data points to be true data, which does not appear to be the case. The authors should reconsider how they present this data, and perhaps offer more insight into the specific number of times each age group was weighed, and with how things are represented it’s possible that some 28-month-old mice weighed only weighed a few times, which is not trustworthy.

We thank the reviewer for the comment. First, we would like to clarify that the AHMM model does not predict or impute missing data. Instead AHMM uses all the temporal data from each mouse without requiring that all the mice have the same number of weight measurements. When we mention “50% of animals were weighed over 10 times, whereas 17% were weighted less than 5”, this refers to the number of weight measurements that could span different months of age. 

 While our colony was not preplanned to have equal measurements for all the animals, we recognize that such an ideal condition is rarely if ever met in most academic research laboratories. Because most research laboratories assume homogeneity of aging in their colony and conduct experiments based on this assumption, we tested the hypothesis of homogeneity of aging in our own colony through a retrospective analysis of fundamental properties of aging such as change in weight. 

The AHMM is used to capture the most probable weight trajectories underlying all real longitudinal weight measurements of the colony.

This study is a proof of concept that aims to demonstrate that a retrospective analysis of an existing aging mouse colony can provide information relevant to age-related phenotypes. In fact, it is often assumed that mice colony are homogeneous, however, our findings indicate that mice in our colony are heterogeneous with different weight aging trajectories that could influence both mechanistic and translational outcomes. 

3) There is no section 2.4.4 in the methods.

We thank the reviewer for noticing this error. We removed section numbering and modified the manuscript structure according with the journal requirements.

4) The % of each APOE genotype varies slightly between Fig 1A and the results section.

We thank the reviewer for pointing out this discrepancy. The difference was due to a rounding issue, which has been corrected. We have updated the results section to ensure that the percentages are now consistent with Fig 1A.

Line 273: “30% hAPOE3, 28% hAPOE3/4, and 42% hAPOE4/4”

5) In Line 277, the red squares should be referenced to figure 3A rather than 2A.

We thank the reviewer for noticing the reference error and modified accordingly.

Line 285: (red squares in Fig 3a and loop arrows in Fig 3b).

6) In lines 408-409 the total sum of percentages is 83%, leading to doubts regarding the missing 17%. As the first part of the discussion, the authors should mention what this missing % is referring to.

We thank the reviewer for highlighting this. The total sum of the percentages is 83% because the identification of weight trajectories is based on the most probable and stable ending states defined by the transition probability matrix (Fig 3A, Supplementary Table 2 reported below). 

In this matrix, diagonal elements represent the probability of reaching a state and remaining in this stable state. States with higher ending probabilities (p>0.3) are selected as ending states with low likelihood of transitioning to another state. The identified states with this property were B, C, I, F, and J, while states A, D, E, G, and H have lower probabilities of being ending states (see elements highlighted in yellow in the table reported below), and therefore are not stable.

Consequently, the 17% of mice not classified into these most probable ending states are mice assigned to be less probable or not stable states. We reported in Fig S2, the weights of these mice colored according to the last not stable state assigned by AHMM. These mice might correspond to outliers or mice assigned to intermediate or initial states. For example, mice in State H will move with high probability (0.629 probability – colored in blue) to ending State C, whereas the probability of remaining in state A is very low (0.016 probability – colored in yellow). 

We modified the result section accordingly:

Results Lines 307-310: “Note that, not all the mice can be classified in one of the stable end trajectories, 17% (215) of mice in our colony did not have a stable end state and were not further explored. These mice correspond to outliers or mice assigned to intermediate or initial unstable states.”

Currently, the study of intermediate states is out of the scope of this study, as we focused on a proof of concept if emerging phenotypes existed in our colony. To this point we included to the Discussion section the study of intermediate states into our future directions:

Discussion Lines 525-527 “Furthermore, future studies will aim to expand the analysis to a larger colony, allowing for more comprehensive phenotyping of weight trajectories, with stratification based on intermediate states, in addition to their final trajectories.”

Probability of Transitioning From State Probability of Transitioning To State

 A B C D E F G H I J

A 0.016 0.269 0.011 0.001 0.011 0.303 0 0.04 0 0.341

B 0.001 0.657 0 0.025 0 0 0 0.013 0 0.302

C 0.034 0 0.56 0.036 0.001 0.008 0.001 0.293 0.054 0.008

D 0 0.318 0.177 0.072 0.06 0.054 0.003 0.189 0.123 0

E 0 0.045 0.031 0 0 0.656 0.01 0 0.181 0.074

F 0 0.095 0.001 0.028 0.018 0.795 0 0 0.055 0.006

G 0.224 0.007 0.069 0 0.277 0.194 0 0.121 0.065 0.039

H 0.012 0.008 0.629 0 0.029 0 0.004 0.119 0.036 0.159

I 0 0.002 0.001 0.054 0.049 0.433 0 0.02 0.439 0

J 0.001 0.065 0 0.008 0 0.001 0 0.027 0.004 0.889

Figure S2 

Reviewer #2: 

“This study reports weight changes that occur in the humanized APOE transgenic mouse models, ones oftentimes used to model AD due to APOE4 being the strongest risk factor for AD. The authors used Autoregressive Hidden Markov Model (AHMM) to predict weight trajectories of the hAPOE ageing mice of both sexes, demonstrating that there are distinct weight, resilience and vulnerability trajectories in these mice models as there are in human population. Even though the study is statistically sound and well rounded, I suggest the following points to be re-evaluated:

1. The study is based on the effect of the APOE risk factor on the weight trajectories of the ageing mice, however not much if anything has been directly mentioned about the weight trajectories of human APOE4 carriers. This has been previously reported by Ukraintseva, et al., 2024; Holmes et al., 2024; Bell et al., 2017; Backman et al., 2015 and others. I would therefore suggest elaborating more on weight trajectories in human APOE4 carriers and linking them with the results obtained in this study.”

2. As a continuation of the 1st suggestion, it has been shown by Holmes and colleagues (2024) that "APOE4 carriers have 19%–22% (TE p = 0.020–0.039) lower chances of surviving to age 85 and beyond, in part, because they reach peak values of weight at younger ages, and their weight declines faster afterward compared to non-carriers." Such weight trajectory would correspond to trajectories C and I detected in this study, as authors also mentioned the link between these trajectories and prodromal weight decline prior to AD diagnosis (line 429). However, the percentage of AP

---

## [Decision Letter · Decision Letter 1]

6 Nov 2024

Weight Trajectories in Aging Humanized APOE Mice with Translational Validity to Human Alzheimer's Risk Population

PONE-D-24-34922R1

Dear Dr. Vitali

We’re pleased to inform you that your manuscript has been judged scientifically suitable for publication and will be formally accepted for publication once it meets all outstanding technical requirements.

Kind regards,

Maud Gratuze

Academic Editor

PLOS ONE

Additional Editor Comments (optional):

Reviewers' comments:

Reviewer's Responses to Questions

**Comments to the Author**

1. If the authors have adequately addressed your comments raised in a previous round of review and you feel that this manuscript is now acceptable for publication, you may indicate that here to bypass the “Comments to the Author” section, enter your conflict of interest statement in the “Confidential to Editor” section, and submit your "Accept" recommendation.

Reviewer #1: All comments have been addressed

Reviewer #2: All comments have been addressed

2. Is the manuscript technically sound, and do the data support the conclusions?

Reviewer #1: Yes

Reviewer #2: Yes

3. Has the statistical analysis been performed appropriately and rigorously? 

Reviewer #1: Yes

Reviewer #2: Yes

4. Have the authors made all data underlying the findings in their manuscript fully available?

Reviewer #1: Yes

Reviewer #2: Yes

5. Is the manuscript presented in an intelligible fashion and written in standard English?

Reviewer #1: Yes

Reviewer #2: Yes

6. Review Comments to the Author

Reviewer #1: All the comments issued in the original draft have been correctly accordingly. I believe the manuscript has been improved substantially, and would like to offer my congratulations to the authors for the hard work.

Reviewer #2: The authors of the "Weight Trajectories in Aging Humanized APOE Mice with Translational Validity

to Human Alzheimer's Risk Population" paper have successfully addressed the comments raised by the reviewers. More precisely, the suggested studies have been included in the revised discussion, thoroughly explaining their connection with the results observed by the authors. By doing so the authors highlighted the correlation between their main observations on hAPOE mouse models and their relevance and connection to human subjects. My recommendation is therefore to accept the manuscript by Vitali et al. without further revision.

7. PLOS authors have the option to publish the peer review history of their article (what does this mean?). If published, this will include your full peer review and any attached files.

Reviewer #1: No

Reviewer #2: No

---

## [Editor Report · Acceptance letter]

13 Nov 2024

PONE-D-24-34922R1 

PLOS ONE

Dear Dr. Vitali, 

I'm pleased to inform you that your manuscript has been deemed suitable for publication in PLOS ONE. Congratulations! Your manuscript is now being handed over to our production team.

Kind regards, 

on behalf of

Dr. Maud Gratuze 

Academic Editor

PLOS ONE